# Metal–Site Dispersed Zinc–Chromium Oxide Derived from Chromate–Intercalated Layered Hydroxide for Highly Selective Propane Dehydrogenation

**DOI:** 10.3390/molecules29133063

**Published:** 2024-06-27

**Authors:** Lu Xue, Maoqi Pang, Zijian Yuan, Daojin Zhou

**Affiliations:** State Key Laboratory of Chemical Resource Engineering, Beijing University of Chemical Technology, Beijing 100029, China

**Keywords:** layered hydroxide, chromate intercalation, spinel, propane dehydrogenation

## Abstract

Propane dehydrogenation (PDH) is a crucial approach for propylene production. However, commonly used CrO_x_–based catalysts have issues including easy sintering at elevated reaction temperatures and relying on high acidity supports. In this work, we develop a strategy, to strongly anchor and isolate active sites against their commonly observed aggregation during reactions, by taking advantage of the net trap effect in chromate intercalated Zn–Cr layered hydroxides as precursors. Furthermore, the intercalated chromate overcomes the collapse of traditional layered hydroxides during their transformation to metal oxide, thus exposing more available active sites. A joint fine modulation including crystal structure, surface acidity, specific surface area, and active sites dispersion is performed on the final mixed metal oxides for propane dehydrogenation. As a result, Zn_1_Cr_2_–CrO_4_^2−^–MMO delivers attractive propane conversion (~27%) and propylene selectivity (>90%) as compared to other non–noble–metal–based catalysts.

## 1. Introduction

Propylene has a wide range of applications in producing chemicals such as polypropylene, propylene oxide, and isopropyl alcohol [1,2,3,4]. However, traditional production processes, such as steam cracking and fluidized bed catalytic cracking, have low propylene yields, high energy consumption, and high costs [5,6]. Therefore, there is an urgent need to develop efficient propylene production technologies. Propane direct dehydrogenation (PDH), which relies on noble metal Pt–based and CrO_x_–based catalysts, provides a more economic and environmentally friendly route to propylene synthesis than conventional processes [7,8]. CrO_x_–catalysts have greater cost advantages over Pt–based catalysts, but the aggregation of the active sites in CrO_x_–based catalysts at elevated reaction temperatures and the deactivation of catalysts due to the high acidity of commonly used supports (e.g., Al_2_O_3_) remain to be solved [9]. Furthermore, the PDH process requires catalysts with a relatively high surface area to make full use of active sites and manage heat transfer to avoid side–reactions, while commonly prepared metal oxides for PDH suffer from low surface area, especially those annealed at elevated temperature to obtain high mechanical strength. Therefore, the design of a CrO_x_–based catalyst with high surface area, accompanying with stably dispersed active sites and the fine modulation of catalyst acidity, is desperately needed to improve the performance of PDH.

Layered hydroxides are a class of two–dimensional (2D) nanosheet materials, whose laminates are constructed by M^2+^ and M^3+^ hydroxides [10,11]. The net trap effect in layered hydroxide laminates is promising to stably anchor each metal cation during high temperature treatment. As a result, mixed metal oxides (MMOs) derived from layered hydroxides can immobilize the elemental distribution and avoid site aggregation, addressing one major problem in PDH catalyst synthesis [12]. Furthermore, exchangeable anions, for example, chromate in the interlayer of layered hydroxides, can firstly contribute to more available PDH active sites; secondly, chromates do not decompose at elevated temperatures, thus preserving the structural integrity of the MMO and maintaining a larger surface area. This overcomes the issues of structural collapse and surface area reduction typically linked to the use of carbonate–intercalated hydroxide precursors. [13]. Third, the tunable interlayer anions and lamellar cations offer significant potential for regulating the surface acidity of MMOs [14].

Following the merits discussed above, we prepared catalysts for propane dehydrogenation using chromate (CrO_4_^2−^) intercalated Zn–Cr layered hydroxides (ZnCr–CrO_4_^2−^) as precursors. Chromate intercalation enables the formation of structurally stable zinc–chromium mixed metal oxides (ZnCr–MMO) with more dispersed active sites after the calcination of the layered hydroxides, which maintain a high surface area (70.80 m^2^/g) and are structurally stable even at a reaction temperature of 550 °C. In addition, chromate intercalation also contributes to the modulation of surface acidity. Joint efforts contribute to high propane conversion (~27%) and selectivity (>90%) at WHSV = 8.9 h^−1^ of Zn_1_Cr_2_–CrO_4_^2−^–MMO.

## 2. Results and Discussion

### 2.1. Characterization of Catalyst Precursors

We used a co–precipitation method to synthesize precursor layered hydroxides, followed by a calcination at 600 °C for 4 h under air atmosphere to obtain catalysts (Figure 1).

The synthesized precursors were first characterized by XRD to study the effect of chromate intercalation on the crystal structure of layered hydroxides. As shown in Figure 2a, both Zn_1_Cr_2_–CO_3_^2−^ and Zn_1_Cr_2_–CrO_4_^2−^ have layered hydroxide structures. The diffraction peaks of Zn_1_Cr_2_–CO_3_^2−^ at 12.2° and 24.6° correspond to (003) and (006) diffraction peaks of layered hydroxides, respectively. Furthermore, the (003) and (006) diffraction peaks of Zn_1_Cr_2_–CrO_4_^2−^ shift towards lower angles, specifically at 10.9° and 20.1°, respectively, compared to Zn_1_Cr_2_–CO_3_^2−^. According to Bragg’s law, this result can be attributed to the difference in anion radii in the interlayer and provides initial evidence of successful chromate intercalation. The structure is analyzed via Fourier transform infrared (FT–IR) within a range of 4000 to 400 cm^−1^. As shown in Figure 2b, the strong broad absorption band of the layered hydroxide at about 3500 cm^−1^ corresponds to the O–H stretching vibrations of both metal hydroxyl (M–OH) group and hydroxyl group from water molecule in the interlayer. The weak absorption near 1639 cm^−1^ is usually considered the bending vibration of the interlayer or physically adsorbed water [15]. Absorption bands at 1365 cm^−1^ and 1492 cm^−1^ for Zn_1_Cr_2_–CO_3_^2−^ originate from C–O stretching vibrations in carbonates [16], whereas no obvious characteristic peaks belonging to CO_3_^2−^ are observed in the chromate–intercalated precursor in the range of 1365–1500 cm^−1^, and absorption bands originating from the Cr(VI)–O vibration appear at about 900 cm^−1^ in the grey box. The Cr(VI)–O vibrational bands of Zn_1_Cr_2_–CrO_4_^2−^ are shifted compared to potassium chromate (887 cm^−1^), which can be attributed to the van der Waals forces between the intercalated chromate and the laminate (also known as confinement effect). Here, we can draw the conclusion of the successful intercalation of chromate.

The morphology of materials was studied by transmission electron microscopy (TEM). As shown as Appendix A, a similar nanosheet morphology between Zn_1_Cr_2_–CrO_4_^2−^ and Zn_1_Cr_2_–CO_3_^2−^ indicates that the intercalation of chromate does not affect the morphology of layered hydroxides. Thermal stability was assessed using thermogravimetric analysis (TGA). As shown in Figure 2c,d, most of the weight loss is completed before 400 °C. The Zn_1_Cr_2_–CO_3_^2−^ presents three weight loss peaks: the first peak (108 °C) indicates the removal of water, the second peak (240 °C) reflects the elimination of interlayer hydroxyl and interlayer anions (CO_3_^2−^), accompanied by the collapse of the layered structure, and the third peak (375 °C) corresponds to the decomposition of CO_3_^2−^ and the elimination of hydroxyl groups [17]. Similar three weight loss peaks were also observed for the Zn_1_Cr_2_–CrO_4_^2−^ precursor, but they were located at different temperatures. The first one at 114 °C is due to removal of surface adsorbed water, and the other two weight loss peaks (228 °C, 392 °C) represents the removal of interlayer hydroxyl groups. The chromate in the Zn_1_Cr_2_–CrO_4_^2−^ interlayer did not thermally decompose, and the overall weight loss was less than that of the Zn_1_Cr_2_–CO_3_^2−^. Results here suggest that Zn_1_Cr_2_–CrO_4_^2−^ has a potential high surface area since decomposition of carbonate will result in structural collapse, which is not the case in Zn_1_Cr_2_–CrO_4_^2−^.

Subsequently, the surface structure and chemical state of the materials were analyzed using Raman spectra and X–ray photoelectron spectra (XPS). As shown in Appendix A, for Zn_1_Cr_2_–CO_3_^2−^ precursors, the strong peak at 536 cm^−1^ is attributed to Cr–O–H bending modes [18,19], while the strong peak of Zn_1_Cr_2_–CrO_4_^2−^ at 850 cm^−1^ is attributed to chromate (CrO_4_^2−^) [20]. Cr 2p spectrum of Zn_1_Cr_2_–CrO_4_^2−^ in Appendix A displays two different valence states. The binding energies of Cr^3+^ in hydroxide are located at ~578 eV (2p^3/2^) and ~588 eV (2p^1/2^) [21], while the binding energy of Cr^6+^ in chromate is located at ~580 eV (2p^3/2^) and ~590 eV (2p^1/2^) [21].

### 2.2. Characterization of Spinel Formation after Precursor Calcination

The prepared layered hydroxide precursors were calcined at 600 °C for 4 h under air atmosphere to obtain mixed metal oxides. We firstly performed structural analysis of the calcined samples using XRD. As shown in Figure 3a, the diffraction peaks of Zn_1_Cr_2_–CrO_4_^2−^–MMO correspond well to ZnCr_2_O_4_ spinel phase (PDF#22–1107), while the diffraction peak attributed to ZnO (PDF#36–1451) is observed at 31° for Zn_1_Cr_2_–CO_3_^2−^–MMO. ZnO is unstable under propane dehydrogenation reaction conditions and can be easily reduced to metallic zinc (Zn) with a melting point of approximately 420 °C [22]. To investigate the effect of Zn/Cr ratios on the structure of the catalysts, we modulated Zn/Cr ratios and calcined each catalyst in the same way. XRD analysis in Appendix A exhibits both ZnO and ZnCr_2_O_4_ phases in samples when Zn/Cr > 0.5, while the diffraction peak of Cr_2_O_3_ was observed in samples where Zn/Cr < 0.5 (PDF#38–1479), which can be explained by the unbalanced A(II)/B(III) ratio in the precursor to form a pure phase spinel. In order to form a pure phase spinel structure, we chose a sample with a Zn/Cr ratio of 1:2 for further study. We calcined Zn_1_Cr_2_–CrO_4_^2−^ at 500–800 °C to explore the effect of calcination temperature on the structure of the material. As shown by the XRD results (Appendix A), the layered structures were destroyed during calcination at temperatures over 500 °C, accompanied by the appearance of ZnCr_2_O_4_ spinel characteristic diffractions, which became sharper along with an elevated calcination temperature from 500 to 800 °C. In addition, ZnO is formed when the temperature is further increased. We finally chose a calcination temperature of 600 °C and a reaction temperature of 550 °C, given the propane dehydrogenation is frequently carried out at 550–700 °C [23].

Appendix A shows the TEM images of the calcined samples and the corresponding size distributions. The particle size of Zn_1_Cr_2_–CO_3_^2−^–MMO is concentrated mainly in the range of 20–25 nm, while the particle size of Zn_1_Cr_2_–CrO_4_^2−^–MMO is in the range of 9–11 nm. Figure 3b,c show the N_2_ isothermal –adsorption–desorption isotherm of the calcined materials. Both Zn_1_Cr_2_–CO_3_^2−^–MMO and Zn_1_Cr_2_–CrO_4_^2−^–MMO exhibit type IV isothermal and H3 hysteresis loops, which are caused by slit–shaped pores formed by the stacking of plate–like particles retained by the precursor [24]. In addition, it can be observed that the specific surface area of Zn_1_Cr_2_–CrO_4_^2−^–MMO (70.80 m^2^/g) is significantly greater than that of Zn_1_Cr_2_–CO_3_^2−^–MMO (12.42 m^2^/g), which is advantageous for active sites exposure.

Raman spectra were used to analyze the changes in the surface structure of the precursor after calcination. The sharp peak at 180 cm^−1^ in Zn_1_Cr_2_–CO_3_^2−^–MMO (Figure 3d) is derived from the Zn–O bending vibration of the ZnCr_2_O_4_ spinel. The band at 500 cm^−1^ with medium intensity and the less intense bands at 599 cm^−1^ can be assigned to the Cr–O stretching vibration (F_2g_). The band at 673 cm^−1^ is attributed to the symmetric Cr–O stretching vibration of A_1g_ symmetry, which originates from the CrO_6_ groups in the spinel structure [25]. The Zn_1_Cr_2_–CrO_4_^2−^–MMO catalyst exhibits a strong peak at 850 cm^−1^ (Figure 3d), which is attributed to the Cr–O stretching vibration of chromate [20], indicating that a significant amount of chromate (CrO_4_^2−^) is present on the catalyst. In addition, chemical state analysis of the calcined samples was performed using XPS. As shown in Figure 3e,f, the binding energies of Cr^3+^ in Cr_2_O_3_ or ZnCr_2_O_4_ spinel are located at ~577 eV (2p^3/2^) and~586.8 eV (2p^1/2^), and the binding energies of Cr^6+^ in CrO_3_ or CrO_4_^2−^ are located at ~579 eV (2p^3/2^) and~588.8 eV (2p^1/2^) [21,26]. The XPS results reveal the presence of both Cr (III) and Cr (VI) in the calcined samples. Additionally, it is found that Zn_1_Cr_2_–CrO_4_^2−^–MMO have a higher proportion of Cr (VI) on the surface. Under reaction conditions, Cr^6+^ can be reduced to Cr^3+^ as active sites to facilitate PDH conversion, while initial Cr^3+^ can be reduced to metallic Cr, a bad active site for PDH [9,27].

### 2.3. The Catalytic Performance of Catalysts

#### 2.3.1. Catalytic Properties in Direct Propane Dehydrogenation

We evaluated propane dehydrogenation performance of Zn_1_Cr_2_–CO_3_^2−^–MMO and ZnCr–CrO_4_^2−^–MMO with different Zn/Cr ratios. The performance of all the catalysts for the PDH reaction was evaluated at 550 °C, 0.1 MPa, C_3_H_8_:N_2_ = 1:4.32, and WHSV = 22.3 h^−1^. Figure 4a,b display the initial conversion and selectivity of various catalysts (Initial conversion and selectivity here refer to data collected 7 min after propane is fed). It is clear that the PDH performance of the catalyst using chromate–intercalated hydroxides as precursors (Zn_1_Cr_2_–CrO_4_^2−^–MMO) is superior to that of the catalyst with non–intercalated hydroxides as precursors (Zn_1_Cr_2_–CO_3_^2−^–MMO) for Zn/Cr=1:2. The Zn_1_Cr_2_–CO_3_^2−^–MMO were primarily composed of the ZnCr_2_O_4_ spinel phase and the crystalline ZnO phase (Figure 3a). Due to the fact that dispersed and unsaturated Cr^3+^ is typically regarded as the active site for propane dehydrogenation, Cr is inactive against propane when it enters the spinel phase [28]. Additionally, the propane conversion on crystalline ZnO is low and unstable under reaction conditions [22]. As a result, Zn_1_Cr_2_–CO_3_^2−^–MMO exhibited low PDH activity, with less than 2% propane conversion. Furthermore, the PDH performance exhibited an increase with the rise in Cr ratio in catalysts with various Zn/Cr ratios. The propane conversion rate was highest (13%) when the Zn/Cr ratio was 1:2 (Zn_1_Cr_3_–CrO_4_^2−^–MMO), and decreased to 6.5% with further increase in the Cr ratio. This could be attributed to the fact that as the amount of chromium increases, the Zn_1_Cr_3_–CrO_4_^2−^–MMO exhibited crystalline Cr_2_O_3_ (Appendix A). Previous studies have shown that the activity of alkane dehydrogenation decreases when the accompanied by the formation of crystalline Cr [29], which was the reason for the reduced PDH conversion performance of the Zn_1_Cr_3_–CrO_4_^2−^–MMO catalysts. Accordingly, it is presumed that the chromate is reduced during the reaction to reactive Cr^3+^, which has a positive effect on the reaction.

In addition, Mg_2_Cr_1_–CrO_4_^2−^–MMO was synthesized and tested for its propane dehydrogenation reactivity. As shown in Appendix A, the catalytic activity of the Mg_2_Cr_1_–CrO_4_^2−^–MMO sample for propane dehydrogenation was not superior to that of the Zn_2_Cr_1_–CrO_4_^2−^–MMO sample. Furthermore, the Zn_1_Cr_2_–CO_3_^2−^–MMO sample exhibited the lowest reactivity. Consequently, in this study, the formation of a pure phase spinel structure as a carrier for the main active site, CrO_4_^2−^ (obtained by intercalation of the precursor), without the interference of oxides with other divalent (inhibitor to PDH reaction) or trivalent ions (changed acidity could also result in catalysts deactivation) is crucial for the propane dehydrogenation reaction.

Appendix A show the conversion and selectivity of different samples throughout prolonged periods of reaction. Although Zn_1_Cr_2_–CrO_4_^2−^–MMO exhibits a relatively high initial PDH conversion and selectivity, it suffers a fast deactivation rate. This could be attributed to the abundance of strongly acidic sites (discussed in the following), which generate a significant amount of carbon during the reaction, covering the active sites.

In addition, the weight hourly space velocity (WHSV) has a significant impact on the catalytic performance of catalysts; therefore, we investigated the impact of various WHSV on the initial conversion and selectivity of Zn_1_Cr_2_–CrO_4_^2−^–MMO. Figure 4c,d show that the catalyst’s propane conversion increases as WHSV decreases, with all catalysts exceeding 90% selectivity. At WHSV = 8.9 h^−1^, the initial conversion of propane reaches approximately 27%, and the selectivity towards propylene is above 90%. Compared to previously studied chromium–based catalysts, such as supported catalysts with binary CrZrO_x_ species (CrZrO_x_/SiO_2_) [30], rod–shaped porous alumina–supported Cr_2_O_3_ catalyst (Cr–Al–800) [31], and CrO_x_ supported on high–silica HZSM–5 [32], our catalyst exhibits a competitive advantage (Appendix A).

#### 2.3.2. Investigation of the Acidic Sites of Catalysts

During the PDH process, acidic sites play a key role in facilitating the cleavage of C–H and C–C bonds (conversion and selectivity), while they also result in the formation of carbon deposits (catalyst stability) [33,34]. Therefore, we modulated the acidic sites on the surface of the materials by adjusting the Zn/Cr ratio and intercalating chromate into the layered hydroxide precursor. To explore the reasons resulting in fast deactivation of the catalyst, we analyzed the surface acidity of the catalyst by NH_3_–TPD. As shown in Figure 5a, Zn_1_Cr_2_–MMO exhibits multiple NH_3_ desorption peaks: <250 °C attributing to weak acidic sites, 250–400 °C correlating to medium–strong acidic sites, and >400 °C corresponding to strong acidic sites. It is widely accepted that the area of the NH_3_–TPD is directly related to the amount of acid sites [35], and Appendix A lists the data after integration; the peaks are for each catalyst. For Zn_1_Cr_2_–CO_3_^2−^–MMO, the integral area was larger at temperatures above 400 °C compared to the range of 50–250 °C, which suggests that it has more strongly acidic sites than weakly acidic sites. The intercalation of CrO_4_^2−^ in the precursor resulted in an increase in the NH_3_–TPD integral area for Zn_1_Cr_2_–CrO_4_^2−^–MMO compared to Zn_1_Cr_2_–CO_3_^2−^–MMO, particularly at 50–250 °C. This suggests an increase in the total acidic sites of the catalyst, particularly the weakly acidic sites.

Also as shown in Appendix A and Appendix A, in the catalysts with different Zn/Cr ratios, the increase in Cr content is accompanied by a rise in the medium and strong acid sites of the catalysts, reaching a maximum at a Zn/Cr ratio of 1:2. At this ratio, the medium acid site content is 0.1396 mmol/g, and the strong acid site content is 0.2503 mmol/g. When the Cr content further increases, the medium and strong acid site content of the catalyst are reduced to 0.0880 mmol/g and 0.0567 mmol/g, respectively. This may be due to the fact that the acidic sites of the catalyst increase with increasing Cr^6+^ concentration, which corresponds to an increase in the cracking capacity of the catalyst, making it more susceptible to carbon formation [36]. However, Zn_1_Cr_3_–CrO_4_^2−^–MMO produces more crystalline Cr_2_O_3_ phases (Appendix A), due to a further increase in Cr content, leading to a decrease in acidic sites.

Further, we employed pyridine infrared (Py–IR) spectroscopy to analyze the surface acidity of the samples. The peaks at 1450, 1490, and 1610 cm^−1^ are typically associated with pyridine coordinated to the Lewis acid (LA) site, while the peaks at 1490 and 1540 cm^−1^ are associated with pyridine coordinated to the Brønsted acid (BA) site [37]. As shown in Figure 5b and Appendix A, the characteristic bands appear primarily at 1450, 1490, and 1615 cm^−1^, proving that the synthesized samples are all mainly Lewis acid sites. Previous studies have shown that Lewis acidic sites can facilitate C–H bond fragmentation, while excessive Bronsted acid sites are more favorable for C–C bond fragmentation [38]. Therefore, the catalysts exhibit high propylene selectivity (>90%).

#### 2.3.3. Catalysts after Reaction

A series of characterizations were performed to investigate changes in the chemical states and structure of the catalysts after the reaction. Figure 5c and Appendix A show the XPS after the reaction of Zn_1_Cr_2_–CrO_4_^2−^–MMO and Zn_1_Cr_2_–CO_3_^2−^–MMO, respectively. It can be observed that after the reaction, Cr^3+^ and Cr^6+^ were detected on the surfaces of both Zn_1_Cr_2_–CrO_4_^2−^−MMO and Zn_1_Cr_2_–CO_3_^2−^–MMO catalysts. However, the proportion of Cr^6+^ notably decreased compared to that before the reaction, particularly in the Zn_1_Cr_2_–CrO_4_^2−^–MMO catalysts. This indicates that Cr^6+^ in Zn_1_Cr_2_–CrO_4_^2−^–MMO was reduced to Cr^3+^ and served as active sites during the propane dehydrogenation reaction, thereby positively influencing the reaction.

The Raman spectra show a weakening of the intensity of the Cr–O vibrational peaks belonging to chromate in the catalysts after reaction (Figure 5d), consistent with the XPS results. This further proves that Cr^6+^ is reduced to Cr^3+^ under the reaction conditions, and that Cr^3+^ may function as an active site. Furthermore, the Raman spectra of the Zn_1_Cr_2_–CrO_4_^2−^–MMO catalyst exhibit clear defects (D) and graphitization (G) peaks at approximately 1500 cm^−1^ [39], indicating the presence of carbon deposits on the catalyst. The weak peaks of carbon in the Raman spectrum of Zn_1_Cr_2_–CO_3_^2−^–MMO may be attributed to its low conversion ratio (<2%), which makes the amount of carbon deposits low.

To investigate evolution in catalyst structure after the reaction and the effect of carbon formation on structure and PDH activity, we further characterized the catalyst’s structure and morphology. The structure of the samples after reaction were characterized by XRD, and Appendix A shows that in addition to the preservation of the ZnCr_2_O_4_ structure after the reaction, clear diffraction peaks belonging to carbon were found in Zn_1_Cr_2_–CrO_4_^2−^–MMO. A comparable trend was observed for Zn_1_Cr_2_–CO_3_^2−^–MMO, although the diffraction peaks for carbon were weaker. The findings were consistent with those obtained through Raman spectroscopy.

TEM was used to analyze the morphology of the reacted catalyst. As shown in Appendix A, after the reaction, the particle size of Zn_1_Cr_2_–CO_3_^2−^–MMO samples was concentrated within the range of 35–45 nm, which is significantly larger (20–25 nm) than the particle size before the reaction. In contrast, the particle size of the Zn_1_Cr_2_–CrO_4_^2−^–MMO catalyst after the reaction was mainly concentrated at 10–11 nm, which is similar to the particle size before the reaction (9–11 nm). It is suggested that Zn_1_Cr_2_–CrO_4_^2−^–MMO is structurally more stable than Zn_1_Cr_2_–CO_3_^2−^–MMO and can reduce the risk of active site aggregation during propane dehydrogenation reaction conditions. In addition, Zn_1_Cr_2_–CO_3_^2−^–MMO and Zn_1_Cr_2_–CrO_4_^2−^–MMO showed black shadows on their surfaces after the reaction, which could be attributed to the presence of carbon deposits on the catalyst surface. At the same time, carbon deposits block the pore structure of the catalyst, further reducing the BET specific surface area; the specific surface areas of Zn_1_Cr_2_–CO_3_^2−^–MMO and Zn_1_Cr_2_–CrO_4_^2−^–MMO were reduced to 10.43 and 49.42 m^2^, respectively. (Figure 5e,f).

Subsequently, we further investigated the carbon deposits after the sample reaction by analyzing the reacted samples using thermogravimetric analysis (TGA). Appendix A shows that the main weight loss of Zn_1_Cr_2_–CrO_4_^2−^–MMO occurs around 300 °C, indicating a substantial accumulation of carbon in the catalyst. This is consistent with our discussion of acidic sites above, i.e., although Zn_1_Cr_2_–CrO_4_^2−^–MMO has more weakly acidic sites that are favorable for the reaction, it also has a large number of strongly acidic sites that will inevitably lead to the formation of carbon deposits. A significant accumulation of carbon can cover the active sites, resulting in a decrease in catalytic activity. For the Zn_1_Cr_2_–CO_3_^2−^–MMO, the low weight loss is due to its low conversion rate (<2%), resulting in less carbon formation after the reaction.

## 3. Materials and Methods

### 3.1. Materials and Reagents

All chemical raw materials were analytical reagent grade. Zinc chloride (ZnCl_2_, anhydrous) and potassium chromate (K_2_CrO_4_) were procured from Shanghai Macklin Biochemical Technology Co., Shanghai, China. Sodium hydroxide (NaOH) and sodium carbonate (Na_2_CO_3_) were procured from Sinopharm Chemical Reagent Co., Ltd., Beijing, China. Chromic Chloride Hexahydrate (CrCl_3_·6H_2_O) was procured from West Asia Chemical Technology, Shandong) Co., Ltd., Shandong, China). 

### 3.2. Catalyst Preparation

The co–precipitation method has been used for the preparation of CrO_4_^2−^ intercalated layered hydroxide precursors (ZnCr–CrO_4_^2−^ and Mg_2_Cr_1_–CrO_4_^2−^). To prevent the intercalation of carbonate into the hydroxide interlayer, nitrogen–saturated de–carbonated water was used instead of deionized water in the subsequent steps. In the case of Zn_1_Cr_2_–CrO_4_^2−^, for example, 4 mmol ZnCl_2_ (anhydrous) and 8 mmol CrCl_3_·6H_2_O were dissolved in 50 mL of deionized water, which was labeled as Solution A; 50 mL of deionized water were used to dissolve 15 mmol NaOH and 10 mmol K_2_CrO_4_. The resulting solution was labeled as solution B. The C solution was prepared by dissolving 20 mmol of K_2_CrO_4_ in 50 mL of deionized water. Solutions A and B were slowly added simultaneously under nitrogen protection to solution C with vigorous stirring at 50 °C for 1.5 h while maintaining a controlled pH around 9. The resulting precipitates were thoroughly washed with deionized water until pH neutrality was reached and then freeze–dried overnight for further use.

The synthesis of Zn_1_Cr_2_–CO_3_^2−^, Zn(OH)_2_, and Cr(OH)_3_ samples was conducted using the co–precipitation method. The synthesis of Zn_1_Cr_2_–CO_3_^2−^ is presented here as an illustrative example. 4 mmol ZnCl_2_ (anhydrous) and 8 mmol CrCl_3_·6H_2_O were dissolved in 50 mL of deionized water, which was labeled as Solution A; 50 mL of deionized water were used to dissolve 15 mmol NaOH and 6 mmol Na_2_CO_3_ (anhydrous), and the resulting solution was labeled as solution B. Solutions A and B were slowly added simultaneously to 50 mL of deionized water with vigorous stirring at 50 °C for 1.5 h while maintaining a controlled pH around 9. The resulting precipitates were thoroughly washed with deionized water until pH neutrality was reached and then freeze–dried overnight for further use.

The prepared layered hydroxide precursors were placed in a muffle furnace and calcined at a temperature of 600 °C for 4 h at a ramp rate of 10 °C/min. Comparison samples were also prepared by calcining at 500, 700, and 800 °C for 4 h at the same ramp rate.

### 3.3. Characterization

The Shimadzu XRD–6000 diffractometer(Shimadzum, Kyoto, Japan) was used to analyze the crystalline structures of all samples, using a Cu Kα source (λ = 0.154 nm) and a scanning range of 5 °C to 90 °C with a scanning step of 10 °C/min. The instrument used for XPS characterization was a ESCALAB 250 (Thermo Fisher Scientific, Waltham, MA, USA), and the X–ray source was an Al Kα target (hv = 1486.6 eV). FT–IR spectra were recorded on a Bruker Vector–22 Fourier Transform spectrometer (Bruker Optics, Karlsruhe, Germany) with 2 cm^−1^ resolution in the 4000 to 400 cm^−1^ range. The Raman spectra were recorded using a 785 nm laser light source (InVia Reflex, Renishaw, London, UK), covering a range of 4000 to 100 cm^−1^. The transmission electron microscopy (TEM) using HT7700 (Hitachi, Tokyo, Japan). The specific surface area was determined using the Brunauer–Emmett–Teller (BET) method, which is based on the adsorption isotherm (ASAP–2460–4N, Micromeritics, Wikiwand, GA, USA). Prior to measurement, the samples were degassed at 180 °C for 2 h. Thermogravimetric analysis (TGA) was used to determine the thermal properties of the samples in air. The weight loss was measured using TGA/DSC 3+ (Mettler Toledo, Zurich, Switzerland) with a heating rate of 10 °C min^−1^. NH_3_ temperature–programmed desorption (NH_3_–TPD) was conducted on the Micrometric ChemiSorb 2750 (Micromeritics, Wikiwand, GA, USA). Following pretreatment, the sample was exposed to the probe molecule (NH_3_) at a flow rate of 40 mL min^−1^ for 1 h to ensure maximum adsorption. Subsequently, data were recorded in He from room temperature to 800 °C at a rate of 10 °C min^−1^. Pyridine–infrared (Py–IR) was conducted on the Bruker Tensor 27 (Bruker Optics, Karlsruhe, Germany). The samples were pressed from the support sheet, placed in the in situ cell of the infrared spectrometer, sealed, evacuated at 350 °C, held for 1 h, and then cooled to room temperature. The in situ cell was filled with bare yard vapors and left to adsorb for 1 h. Afterward, the cell was evacuated and heated to 200 °C and 350 °C, respectively, and degassed for 1 h at each temperature. Bare–infrared spectra were collected at both 200 °C and 350 °C.

### 3.4. Catalytic Test

The reactor was heated to a reaction temperature of 550 °C with a ramp rate of 10 °C/min using 44.2 mL min^−1^ N_2_. After reaching the reaction temperature, the feed composition was changed to 10.22 mL min^−1^ C_3_H_8_ and 44.2 mL min^−1^ N_2_ to measure the performance of the catalyst. At a constant feed composition, catalyst weights of 0.2 g, 0.4 g, and 0.5 g were employed to modulate the weight hourly space velocity (WHSV).

The reaction products were analyzed using the GC 2014 online system (Shimadzu, Kyoto, Japan) equipped with an FID detector for hydrocarbons and a TCD detector for hydrogen. Propane conversion (conv. C_3_H_8_) and propylene selectivity (sel. C_3_H_6_) were determined utilizing the internal standard method as per the following equation:(1)$$\mathrm{Conv}.(C_{3}H_{8})(\%)=\frac{F_{{C3H8}_{\mathrm{inlet}}}-F_{{C3H8}_{\mathrm{outlet}}}}{F_{{C3H8}_{\mathrm{inlet}}}}\times 100\%$$
(2)$$\mathrm{Sele}.(C_{3}H_{6})(\%)=\frac{F_{{C3H6}_{\mathrm{outlet}}}}{\sum i\left(\frac{n_{i}}{3}\right)\times F_{i_{\mathrm{outlet}}}}\times 100\%$$

## 4. Conclusions

In this study, ZnCr_2_O_4_ was synthesized through calcination at 600 °C with chromate interaction into a layered hydroxide interlayer as a precursor under air atmosphere. The positive effect of chromate interaction on the propane dehydrogenation reaction allowed the catalyst to form pure phase spinel after calcination at 600 °C, without the interference of the ZnO and Cr_2_O_3_ phases. Furthermore, intercalated chromate helped to maintain large specific surface area and expose active sites; jointly, Zn_1_Cr_2_–CrO_4_^2−^–MMO demonstrated a high propane dehydrogenation activity, achieving 27% propane conversion at WHSV = 8.9 h^−1^ while maintaining selectivity above 90%. Our findings suggest that MMOs prepared with oxometallate intercalated hydroxides as precursors have the potential to be applied to various reactions requiring high surface area and stable active sites against aggregation during reaction.

## Figures and Tables

**Figure 1 molecules-29-03063-f001:**
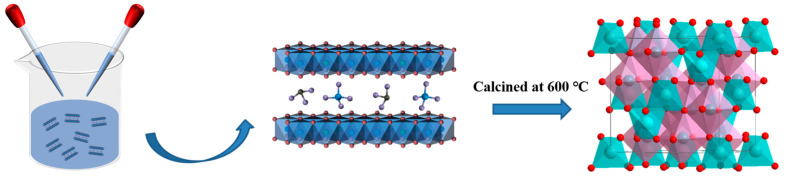
Schematic illustration of the pure ZnCr_2_O_4_ spinel preparation from chromate–intercalated precursor.

**Figure 2 molecules-29-03063-f002:**
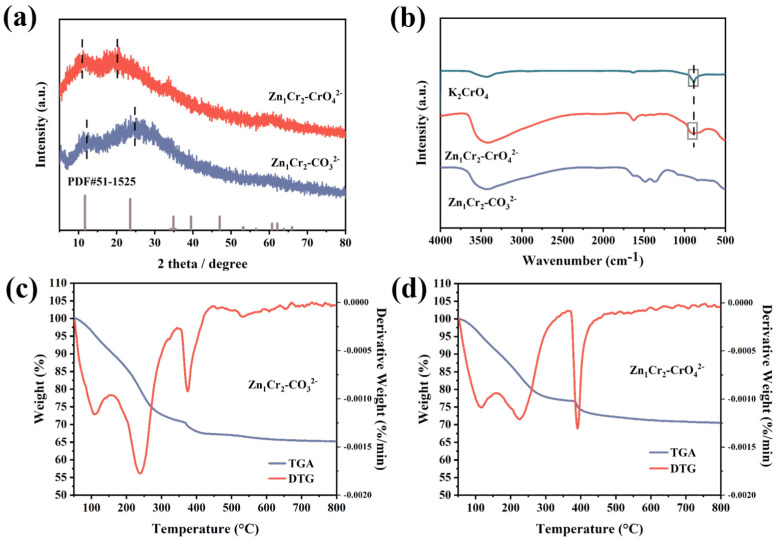
(**a**) XRD patterns, (**b**) FTIR spectrum, (**c**,**d**) TGA–DTA of the Zn_1_Cr_2_–CO_3_^2−^ and Zn_1_Cr_2_–CrO_4_^2−^ precursors.

**Figure 3 molecules-29-03063-f003:**
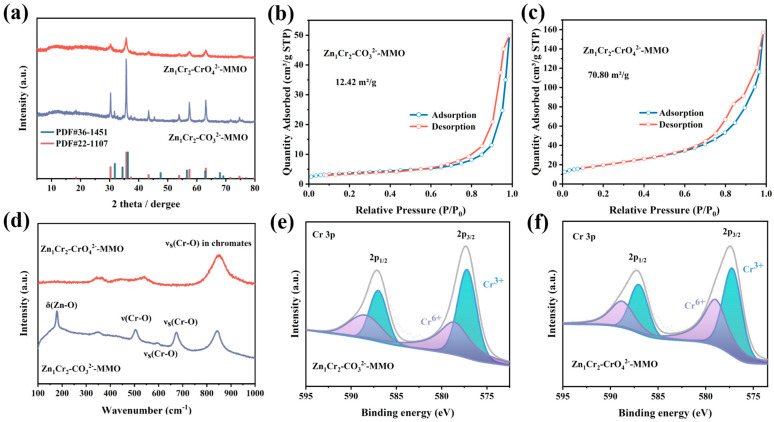
(**a**) XRD patterns of Zn_1_Cr_2_–CO_3_^2−^–MMO and Zn_1_Cr_2_–CrO_4_^2−^–MMO. N_2_ –adsorption–desorption isotherm and pore size distribution of Zn_1_Cr_2_–CO_3_^2−^–MMO (**b**) and Zn_1_Cr_2_–CrO_4_^2−^–MMO (**c**). (**d**) Raman of Zn_1_Cr_2_–CO_3_^2−^–MMO and Zn_1_Cr_2_–CrO_4_^2−^–MMO. XPS spectra of Cr 2p over Zn_1_Cr_2_–CO_3_^2−^–MMO (**e**) and Zn_1_Cr_2_–CrO_4_^2−^–MMO (**f**).

**Figure 4 molecules-29-03063-f004:**
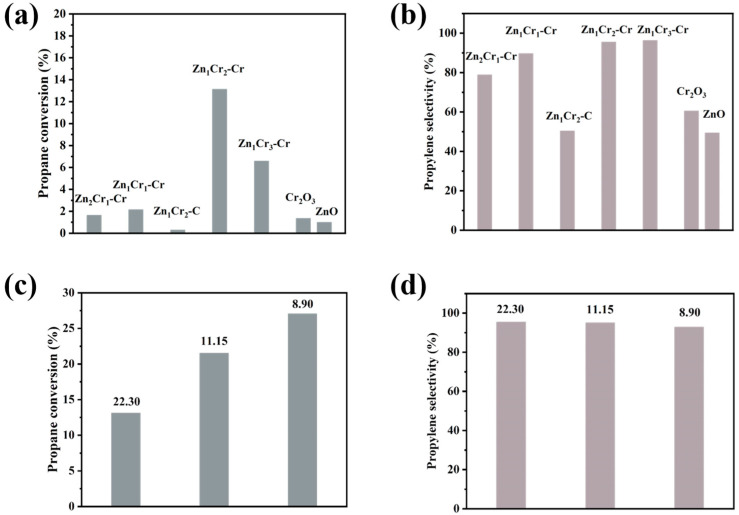
(**a**) Initial propane conversion and (**b**) propylene selectivity of Zn_1_Cr_2_–CO_3_^2−^–MMO and Zn_1_Cr_2_–CrO_4_^2−^–MMO (Reaction conditions: 550 °C, WHSV = 22.3 h^−1^, C_3_H_8_:N_2_ = 1:4.32). (**c**) Initial propane conversion and (**d**) propylene selectivity of Zn_1_Cr_2_–CrO_4_^2−^–MMO with different WHSV (22.30 h^−1^, 11.15 h^−1^ and 8.9 h^−1^).

**Figure 5 molecules-29-03063-f005:**
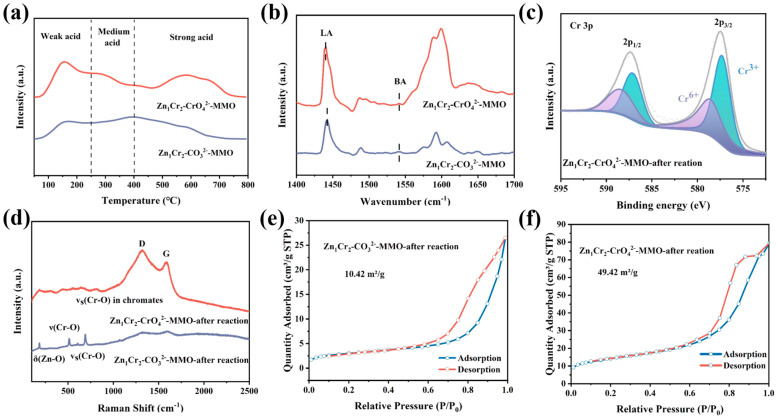
(**a**) NH_3_–TPD, (**b**) Py–IR of Zn_1_Cr_2_–CO_3_^2−^–MMO and Zn_1_Cr_2_–CrO_4_^2−^–MMO. (**c**) XPS spectra of Cr 2p over Zn_1_Cr_2_–CrO_4_^2−^–MMO after reaction. (**d**) Raman spectra of Zn_1_Cr_2_–CO_3_^2−^–MMO and Zn_1_Cr_2_–CrO_4_^2−^–MMO after reaction., N_2_ –adsorption–desorption isotherm and pore size distribution of Zn_1_Cr_2_–CO_3_^2−^–MMO (**e**) and Zn_1_Cr_2_–CrO_4_^2−^–MMO after reaction (**f**).

## Data Availability

The data presented in this study are available in the Appendix A.

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
