# Peer review of "Metal–Site Dispersed Zinc–Chromium Oxide Derived from Chromate–Intercalated Layered Hydroxide for Highly Selective Propane Dehydrogenation"

_molecules, 2024, doi:10.3390/molecules29133063_

Round 1

Reviewer 1 Report

Comments and Suggestions for Authors

This manuscript by Xue et al. was dealt with direct propane hydrogenation(PDH) to propylene based on the designed ZnCr-CrO42--MMO and ZnCr-CO32--MMO catalysts. And their reactivity was comparatively tested and the involved structure characterization before and after testing was employed. Here, I have some considerations listed as follow:

1. As the authors mentioned, the stable ZnCr-MMO structure can avoid catalyst sintering, which was confirmed by the characterization results. However, the forming ZnCr2O4 nearly has no reactivity towards PDH. In this case, the promoting effect of ZnO seemingly lost its significance. Moreover, the addition of ZnO also did not improve the stability of catalyst. In the designed system, whether or not is chromate enough for the completion of PDH? As mentioned, Cr6+ is reduced to Cr3+ under the reaction conditions, and Cr3+ may function as an active site. So, it is advised that the data based on the ZnO-free catalyst should be provided to elaborate the effect of ZnO.

2. From Table S1, the performance of the optimized catalyst in this study is not outstanding compared with the other counterparts.

3. The "initial conversion and selectivity" in Line 180 is a obscure expression, especially for the unstable catalyst. The timespan for the obtained data needs to be specified.

4. What kind of cation neutralize the negative charge of chromates, CrO42-? That is, the forms of chromate in the structure. As mentioned, Cr6+ is reduced to Cr3+ under the reaction conditions. Is there an induction time during the period of PDH?

5. The authors concluded that the forming carbon during the reaction, covering the active sites led to the catalyst deactivation as supported by the Raman and XRD data. Nevertheless, the diffraction peaks due to carbon phase is not obvious and the Raman results did not provided by the authors. Carbon removal via combustion to support the above inference is recommended.

6. In addition, there are still some errors in the English writing, expression and grammar in this manuscript. For example, in Line 9, "Propylene dehydrogenation (PDH) is a crucial approach for propane production"; Line 37, "arear", and Line 44 "inherent"; Chinese characters and a wrong text, "3845–3851 | 3847" have appeared in Table S1. Wrong expression exists in Line 45-52. The authors should spend more time checking for errors in English expression.

Comments on the Quality of English Language

there are still some errors in the English writing, expression and grammar in this manuscript. For example, in Line 9, "Propylene dehydrogenation (PDH) is a crucial approach for propane production"; Line 37, "arear", and Line 44 "inherent"; Chinese characters and a wrong text, "3845–3851 | 3847" have appeared in Table S1. Wrong expression exists in Line 45-52. The authors should spend more time checking for errors in English expression.

Reviewer 2 Report

Comments and Suggestions for Authors

Round 2

Reviewer 1 Report

Comments and Suggestions for Authors

I still have some concerns about the authors' response and Zn1Cr2-CrO42--MMO catalysts. The authors did not fully respond to the reviewer's question 1. So an additional doubt is come into being: The presence of ZnO is inhibitory for propane dehydrogenation as proposed by authors. In this way, can one say that mixed metal oxides (ZnCr-MMO) derived from layered hydroxides just act as a support of CrO42- phase? If so, Whether or not other similar MMOs (rather than ZnCr) also can serve as supports for this active phase? In this case, the negative impact of ZnO can be avoided.

In addition, the first letter of the second to last word in line 71 should be capitalized.

Reviewer 2 Report

Comments and Suggestions for Authors

The manuscript has been improved. The authors have adequately addressed most of my concerns and made the necessary changes to the manuscript. I recommend this paper for publication.
